

# Relationship between salivary flow rate and dental caries in normal and underweight children: a comparative cross-sectional study from district Tharparkar

Abdul Qadir Khan Dall[1], Muhammad Adeel Ahmed[2], Fizza Zulfiqar[3], Sarwat Batool[4], Rizwan Jouhar[2] and Muhammad Faheemuddin[5]

[1] Department of Operative Dentistry, Faculty of Dentistry, Liaquat University of Medical & Health Sciences Jamshoro, Sindh, Pakistan
[2] Department of Restorative Dental Sciences, College of Dentistry, King Faisal University, Al Ahsa, Eastern Province, Saudi Arabia
[3] Department of Oral Medicine, Bhitai Medical and Dental College, Mirpurkhas, Pakistan
[4] Department of Physiology, Liaquat University of Medical & Health Sciences Jamshoro, Sindh, Pakistan
[5] Department of Prosthodontics and Implantology, College of Dentistry, King Faisal University, Al Ahsa, Saudi Arabia

Corresponding author
Muhammad Adeel Ahmed,
mshakeel@kfu.edu.sa

## ABSTRACT

**Background and Objectives:** This study aims to explore the relationship between body mass index (BMI), salivary flow rate, and dental caries among children in Tharparkar.

**Materials and Methods:** A comparative cross-sectional study was conducted over 6 months involving 179 children aged 8–12 years from Tharparkar and Hyderabad. Weight and height were recorded, BMI was computed using height and weight, and salivary flow rate was measured using the spitting method. Dental caries were assessed using the DMFT (decayed, missing, and filled teeth) index. Data were analyzed using SPSS, and independent t-tests were performed to assess group differences.

**Results:** Underweight children (60.9% male, mean BMI 15.46 ± 2.45 kg/m$^2$) showed higher mean DMFT scores for deciduous teeth (2.44 *vs.* 1.06, $p = 0.009$) compared to normal-weight peers. No significant differences were found in permanent teeth DMFT scores or salivary flow rates between BMI groups.

**Conclusions:** Underweight children exhibited a higher prevalence of dental caries in their primary teeth, underscoring the need for integrated nutritional and oral health interventions in vulnerable populations. However, salivary flow rate did not differ significantly between BMI groups, suggesting other factors may play a more critical role in caries prevention.

## INTRODUCTION

Dental caries remains a significant public health issue globally, especially among children. This condition, characterized by the demineralization of tooth enamel due to acids

produced by bacterial activity, is more prevalent in underprivileged regions and populations with limited access to dental care and nutritional resources. District Tharparkar in Pakistan represents one such area, where malnutrition, socio-economic challenges, and a high prevalence of dental caries coalesce, posing unique oral health challenges for children (*Menghwar et al., 2022*).

Saliva plays an indispensable role in maintaining oral health, acting as a natural buffer that neutralizes acids, facilitates food particle clearance, and provides essential minerals for the remineralization of teeth. A sufficient salivary flow rate is critical in protecting against dental caries. However, when the flow rate is reduced, particularly in malnourished or underweight children, the protective functions of saliva are compromised, increasing the risk of caries (*Cunha-Cruz et al., 2013*; *Choudhary et al., 2022*).

The relationship between body weight and dental caries has gained attention, with research suggesting that underweight children may be particularly vulnerable. Malnutrition can impair salivary function, reducing its antibacterial and buffering capabilities, while also affecting the integrity of tooth enamel. Consequently, children who are underweight may not only exhibit lower salivary flow rates, but are also more susceptible to the development of dental caries due to their compromised oral environment (*Menghwar et al., 2022*; *Idrees et al., 2017*; *Paisi et al., 2019*; *Rajendra et al., 2023*).

Despite the known significance of these factors, the correlation between salivary flow rate, body weight, and dental caries remains underexplored, particularly in rural areas like Tharparkar, Sindh, Pakistan. This study aims to address this gap by investigating the relationship between body weight, salivary flow rate, and the prevalence of dental caries in children in this region. By focusing on a vulnerable population, the research seeks to provide insights that could inform public health interventions aimed at reducing caries risk through targeted nutritional and oral health strategies.

## MATERIALS AND METHODS

A comparative cross-sectional study was conducted to examine the relationship between body weight, salivary flow rate, and dental caries in children from district Tharparkar. The study spanned 6 months, from December 14, 2021, to May 15, 2022, and was carried out at Taluka and district hospitals of Tharparkar (Mithi) and Liaquat University Hospital Hyderabad's Advanced Dental Care Center. The ethical approval was obtained from the Research Ethics Committee, Liaquat University of Medical and Health Sciences, Hyderabad, Pakistan (Ref No. LUMHS/REC/-237) on 14 December 2021.

Non-probability consecutive sampling was used to recruit participants within each hospital. A sample size of 174 was initially calculated using Open-Epi Version 3.01, by taking 87% prevalence of dental caries in children (*Sultana et al., 2022*) at a 95% confidence interval and a 5% margin of error.

After obtaining written informed consent from caregivers, children aged 8 to 12 years with at least 10 natural teeth and no medical conditions or medications affecting salivary production were included. The reason for children with at least 10 natural teeth was adequate sampling for assessment of dental caries. Children wearing orthodontic

appliances, those on medication, and those with systemic conditions such as gastroesophageal reflux disease (GERD), cancer, or celiac disease were excluded. Additionally, any child who had received professional fluoride treatment or used antimicrobial oral hygiene products in the past 6 months was also excluded.

Weight and height were measured using a calibrated digital weighing scale (Life Care Medical, Pakistan) and height was measured with a margin of error of 1 cm, with subjects standing barefoot against a wall, feet together, and heads positioned against the wall. A ruler was placed parallel to the ground touching the top of the head. BMI was calculated using the formula weight (kg)/height$^2$ (m) and categorized into underweight (BMI < 18.5), normal weight (BMI 18.5–24.9), over-weight (BMI 25–30), and obese (BMI > 30) (*Weir & Jan, 2023*).

Salivary flow rate was measured after instructing children to refrain from eating, drinking, or practicing oral hygiene for 90 min. The spitting method was employed, where participants chewed on wax to stimulate saliva production (*Priya & Muthu Prathibha, 2017*). After 30 seconds of chewing a piece of wax, saliva was then collected over five minutes, and the flow rate was calculated in milliliters per minute.

Dental caries was assessed using the World Health Organization (WHO) criteria, specifically the decayed, missing, and filled teeth (DMFT) index (*Madhusudhan & Khargekar, 2020*; *Rai et al., 2024*). The participants were examined while seated on a portable dental chair using artificial light with the aid of a plain mouth mirror, periodontal probe, and disposable gauze. A 3-in-1 syringe was used to dry the tooth surface to detect pre-cavitated enamel lesions. Oral examinations were conducted by trained dental professionals to record the DMFT score for each child. Two different examiners noted the dental caries and DMFT scores with an agreement value of more than 90%.

## Statistical analysis

Statistical analysis was performed using SPSS version 24. Data on demographic variables such as age, gender, socio-economic status, family size, and occupation were collected. Categorical variables like gender and BMI were analyzed for frequency and percentage distributions, while continuous variables included age, weight, height, BMI, salivary flow rate. Numerical variables such as DMFT scores were analyzed for means and standard deviations after assessing normal distribution with the Shapiro-Wilk test. Median with inter-quartiles were reported when data was non-normal. Non-normally distributed variables between two groups using the Mann-Whitney U test. Statistical significance was set at $p \leq 0.05$.

## RESULTS

The study included a total of 179 children. The gender distribution showed 60.9% male and 39.1% female with average age of 10.06 ± 1.34 years. The participants had a mean weight of 26.58 ± 6.93 kg, and their average height was 4.28 ± 0.37 feet. The mean BMI of the participants was 15.46 ± 2.45 kg/m$^2$, with a considerable portion of the children categorized as underweight.

**Table 1 Comparison of salivary flow rate, and DMFT scores between underweight (BMI ≤ 18.4) and normal-weight (BMI 18.5–24.9) children.**

| BMI groups | Descriptive statistics | Salivary flow rate | DMFT permanent | DMFT primary |
|---|---|---|---|---|
| Underweight | Median | 4 | 0 | 1 |
| | Q1–Q3 | 3–5 | 0–1 | 0–4 |
| | Mean | 4.1 | 0.70 | 2.4 |
| | Standard deviation | 2.1 | 1.2 | 2.7 |
| Normal weight | Median | 4.5 | 0.5 | 1.1 |
| | Q1–Q3 | 2.3–5 | 0–1.8 | 0–1.8 |
| | Mean | 3.8 | 0.9 | 1.1 |
| | Standard deviation | 1.5 | 1.1 | 1.7 |

**Table 2 Comparison of salivary flow rate, and decayed, missing, and filled teeth (permanent and deciduous) using Mann-Whitney U test.**

| Variables | Groups | Means ranks | Mann-Whitney U | p-value |
|---|---|---|---|---|
| Salivary flow rate of subjects (ml/5 min) | Underweight | 90.10 | 1,288.50 | 0.936 |
| | Normal weight | 89.03 | | |
| Decayed, missing, filling teeth of permanent teeth | Underweight | 88.81 | 1,109.50 | 0.243 |
| | Normal weight | 102.16 | | |
| Decayed, missing, filling teeth of deciduous | Underweight | 92.57 | 885.00 | *0.029 |
| | Normal weight | 63.81 | | |

**Note:**
* Statistically significant.

The mean salivary flow rate was 4.09 ± 2.09 ml/5 min. The mean DMFT score for permanent teeth was 0.71 ± 1.23, reflecting a relatively low prevalence of dental caries in the permanent dentition. In contrast, the mean DMFT score for deciduous teeth was 2.32 ± 2.64, indicating a higher incidence of dental caries in primary teeth among the children studied.

Table 1 presents the median and mean values for salivary flow rate, and DMFT scores for underweight and normal-weight children. Underweight children (BMI ≤ 18.4) had a slightly higher salivary flow rate compared to the normal-weight group (BMI 18.5–24.9). Notably, underweight children also exhibited a higher mean DMFT score for deciduous teeth.

The Mann-Whitney U test revealed significant differences between groups for several variables. For weight, there was a statistically significant difference with lower means ranks in underweight children (Mann-Whitney U = 129.50, $p < 0.001$). For Body Mass Index (BMI), mean rank was 82 for underweight whereas mean rank for normal weight was 163.

In contrast, there was no statistically significant difference in salivary flow rate between groups (U = 1,288.50, $p = 0.936$), and decayed, missing, and filled permanent teeth also showed no significant difference (U = 1,109.50, $p = 0.243$). However, for decayed, missing, and filled deciduous teeth, a significant difference was found (U = 885.00, $p = 0.029$) with mean rank of 82 in underweight, and 171.5 in normal weight children (Table 2).

## DISCUSSION

This study aimed to explore the relationship between body mass index (BMI), salivary flow rate, and dental caries in children from Tharparkar, a region facing significant socio-economic challenges. The results revealed several important findings, particularly regarding the association between BMI and the prevalence of dental caries in primary teeth, as well as insights into salivary flow rate across different BMI groups.

This study found that permanent teeth had lower DMFT scores in both underweight children and normal weight, reflecting a relatively low prevalence of dental caries in the permanent dentition. The lower DMFT score could be due to the fact that the teeth had just erupted. However, subgroup analysis could not be done further to explore the reason for the lower DMFT score in permanent teeth, as data was not recorded regarding the eruption status.

On the other hand, a higher DMFT score for deciduous teeth was observed among underweight children compared to their normal-weight counterparts. This suggests that underweight children are more susceptible to dental caries in their primary dentition, which aligns with existing research, indicating that malnutrition impairs both tooth mineralization and saliva's protective qualities (*Madhusudhan & Khargekar, 2020*). The compromised oral environment in underweight children, particularly due to malnutrition, could explain the heightened vulnerability of deciduous teeth to caries development. Previous studies have also demonstrated similar results, highlighting the critical role that adequate nutrition plays in promoting both dental and general health (*Menghwar et al., 2022*; *Choudhary et al., 2022*; *Idrees et al., 2017*).

However, it is important to note that no significant difference was observed between the BMI groups concerning DMFT scores for permanent teeth. This could be attributed to several factors, including the prolonged exposure of permanent teeth to fluoride from toothpaste and other preventive measures, that may offset the risks posed by poor nutritional status. Additionally, permanent teeth may have a stronger enamel structure and more developed immune responses in older children, making them less vulnerable to the impact of malnutrition compared to primary teeth (*Idrees et al., 2017*; *Andrysiak-Karmińska et al., 2022*; *Bhagavatula et al., 2016*).

Interestingly, the study did not find a significant difference in salivary flow rates between underweight and normal-weight children. This finding contradicts some previous research suggesting that malnutrition can impair salivary gland function, leading to reduced saliva production and increased caries risk (*Bhagavatula et al., 2016*; *Müller et al., 2023*). The lack of variation in salivary flow rate across BMI groups in this study may suggest that other factors, such as the composition and quality of saliva, are more crucial in preventing dental caries rather than just the quantity. The buffering capacity, antimicrobial properties, and mineral content of saliva are all key factors in protecting against caries, and these may be compromised in malnourished children irrespective of the overall salivary flow (*Cunha-Cruz et al., 2013*; *Choudhary et al., 2022*; *Laputková et al., 2018*). Limited sample size could be a factor as well for non-significant association between saliva flow rates and BMI.

The discrepancies between our findings and those of other studies could be explained by differences in environmental factors such as diet (*Yadav et al., 2022*), oral hygiene practices (*Adler et al., 2021*), and access to healthcare (*Hotchandani et al., 2022*). For instance, while some studies have linked both overweight and underweight status to increased caries risk, others have found no clear association between BMI and caries (*Paisi et al., 2019*; *Alshihri et al., 2019*). Regional dietary habits, availability of fluoride in drinking water, and the level of oral hygiene education can all vary significantly between study populations, potentially accounting for the varied findings (*Choudhary et al., 2022*; *Hotchandani et al., 2022*; *Alshihri et al., 2019*).

The findings of this study underscore the importance of addressing malnutrition as a part of oral health interventions in vulnerable populations such as those in Tharparkar. Improving nutritional intake among underweight children could have a direct impact on reducing the risk of dental caries, particularly in primary teeth. Moreover, public health initiatives aimed at enhancing access to fluoride treatments, and promoting better oral hygiene practices should be integrated into broader efforts to improve the overall health and well-being of children in this region.

The study suffers with some limitations. First, we evaluated children on basis of BMI. The study did not account for dietary habits and oral hygiene status due to which findings were not adjusted for these confounding variables. The presentation of two BMI groups was not 1:1. These gaps limit the generalizability of the results to a wider population. Thus, further research is needed to explore the role of saliva composition in caries prevention and to clarify the complex relationship between BMI, salivary flow, and oral health in different populations.

## CONCLUSIONS

This study found that underweight children in Tharparkar have a higher risk of dental caries in their primary teeth, while salivary flow rates did not differ significantly between BMI groups. These findings emphasize the need for integrated nutritional and oral health strategies to reduce caries risk in vulnerable populations.

### Funding

This work was supported by the Deanship of Scientific Research, Vice Presidency for Graduate Studies and Scientific Research, King Faisal University, Saudi Arabia [KFU250447]. The funders had no role in study design, data collection and analysis, decision to publish, or preparation of the manuscript.

### Grant Disclosures

The following grant information was disclosed by the authors:
Deanship of Scientific Research.
Graduate Studies and Scientific Research, King Faisal University, Saudi Arabia [KFU250447].

## Competing Interests

The authors declare that they have no competing interests.

## Author Contributions

- Abdul Qadir Khan Dall conceived and designed the experiments, performed the experiments, prepared figures and/or tables, and approved the final draft.
- Muhammad Adeel Ahmed performed the experiments, analyzed the data, authored or reviewed drafts of the article, and approved the final draft.
- Fizza Zulfiqar conceived and designed the experiments, performed the experiments, prepared figures and/or tables, and approved the final draft.
- Sarwat Batool performed the experiments, prepared figures and/or tables, and approved the final draft.
- Rizwan Jouhar analyzed the data, authored or reviewed drafts of the article, and approved the final draft.
- Muhammad Faheemuddin analyzed the data, authored or reviewed drafts of the article, and approved the final draft.

## Human Ethics

The following information was supplied relating to ethical approvals (*i.e.*, approving body and any reference numbers):

Research Ethics Committee, Liaquat University of Medical and Health Sciences, Hyderabad, Pakistan.

## Data Availability

The original measurements are available in the Supplemental File.

## Supplemental Information

Supplemental information for this article can be found online at http://dx.doi.org/10.7717/peerj.19128#supplemental-information.

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
