# Peer review of "Relationship between salivary flow rate and dental caries in normal and underweight children: a comparative cross-sectional study from district Tharparkar"

_PeerJ, doi:10.7717/peerj.19128_

## Round 0.1 · original submission · Major Revisions

Dear authors,
Your manuscript has been reviewed by three external peer reviewers. It addresses a relevant public health concern. The introduction can be improved by incorporating the reviewers' suggestions. Ethical approval must be explicitly stated, and missing methodological details, such as the spitting method, saliva collection timing, and dietary restrictions, need clarification. Additionally, details on interexaminer calibration and references supporting sample size justification are essential. The discussion should be revised to align more closely with the study's objectives and findings.

·

Basic reporting

Can be improved. Attached comments

Experimental design

Yes. with in the aims. The research question is defined. But the scope for improvement is there. Attached comments

Validity of the findings

Needs to be revised—multiple queires. Comments attached

Reviewer 2 ·

Basic reporting

The manuscript is well-written and uses professional English throughout.

The introduction gives sufficient details, including the link between BMI, flow rate, and caries. However, I would like the others to elaborate a bit more on the relationship between dental caries and nutrition using newer studies.

The figures and tables are clearly labelled and summarise data as necessary. Please look into Table 1 and see if for primary teeth you would use dmft/s or DMFT.

Raw data is not shared, but it is referenced at places.

Ethical approval can be mentioned in line 60.

Experimental design

The research question is clearly stated, addressing a meaningful knowledge gap. The focus on a vulnerable population (children in Tharparkar) adds value.

The comparative cross-sectional design is appropriate for the research aims. Inclusion and exclusion criteria are well-defined, ensuring a focused participant pool. Please provide a reference for the statement that an additional 10% in the sample increases the power of the study. The inclusion criteria mentions children with at least 10 natural teeth, please provide an explanation or reference as to why this number was agreed upon.

The methods for measuring BMI, salivary flow rate, and DMFT scores are sufficiently detailed, enabling replication. How was sampling bias reduced? Please provide reference as to what spitting method was followed and also why stimulated saliva is chosen. Was any particular time chosen for the saliva collection? What about dietary restrictions? were confounding factors like diet and oral hygiene checked upon. Please provide reference on how oral evaluation was performed; it is mentioned based on WHO, provide references please. There is no mention of how interexaminer claibration was done! what was the crinbach value?

Validity of the findings

The statistical methods are appropriate and well-documented.
However, the lack of significant differences in salivary flow rates across BMI groups suggests a need for more nuanced exploration of salivary composition.

How were the confounding factors, like diet and oral hygiene, accounted for? It is not mentioned as an objective nor in the methodology.Discussion needs to align with the research objectives and results

Conclusions are well stated.

Additional comments

None

Reviewer 3 ·

Basic reporting

No Comment

Experimental design

Equal number of male and female subjects could have been taken for uniform distribution within the population
No novelty in the topic, conceptualization or design

Validity of the findings

No comment

Additional comments

In the Consent form, the subheading "Procedure" has a few grammatical errors which should be edited.

---

## Round 0.2 · Minor Revisions

Dear authors,
One of the reviewers provided comments as a minor revision; please address them.

·

Basic reporting

Clear

Experimental design

No comment

Validity of the findings

No comment

Additional comments

Mann-Whitney Y test should be Mann-Whitney U test.
Table 2 doesn't need comparisons of weight, height and BMI among underweight and normal weight children. Kindly remove.
Add mean and SD along with median and Q and Q3.

Reviewer 3 ·

Basic reporting

This has been improved upon and edited and is satisfactory in its present form.

Experimental design

No comment

Validity of the findings

Corrections according to the reviews have been satisfactorily incorporated.

Additional comments

No comment

---

## Round 0.3 · accepted · Accept

Dear authors,
We are pleased to inform you that your manuscript has been accepted for publication in PeerJ. Congratulations on this achievement! We appreciate your valuable contribution to the field and look forward to sharing your work with the scientific community.